# Reduction of Blood Loss by Means of the Cavitron Ultrasonic Surgical Aspirator for Thoracoscopic Salvage Anatomic Lung Resections

**DOI:** 10.3390/cancers15164069

**Published:** 2023-08-11

**Authors:** Yu-Hsiang Wang, Stella Chin-Shaw Tsai, Frank Cheau-Feng Lin

**Affiliations:** 1Department of Thoracic Surgery, Chung Shan Medical University Hospital, Taichung 40201, Taiwan; 99311132tzu@gmail.com; 2Superintendent Office, Tungs’ Taichung MetroHarbor Hospital, Taichung 43503, Taiwan; tsaistella111@gmail.com; 3Department of Post-Baccalaureate Medicine, College of Medicine, National Chung Hsing University, Taichung 40227, Taiwan; 4School of Medicine, Chung Shan Medical University, Taichung 40201, Taiwan

**Keywords:** blood loss, cavitron ultrasonic surgical aspirator (CUSA), lung cancer, salvage surgery, surgical outcomes

## Abstract

**Simple Summary:**

Surgery for centrally located lung cancer after previous cancer treatments can be challenging due to the lungs adhering to the chest wall. We investigated whether using the Cavitron Ultrasonic Surgical Aspirator (CUSA) could help. We focused on patients who had thoracoscopic salvage anatomic lung resections after systemic or radiotherapy with and without CUSA. The CUSA group took more time for surgery but had less bleeding. Our analysis found that older age carried more risk for complications, and using CUSA decreased bleeding, whereas radiotherapy increased the risk of bleeding. CUSA seems to prevent harm to blood vessels and airways, which could lead to less bleeding during this specific type of minimally invasive lung cancer surgery.

**Abstract:**

In centrally located lung tumors, salvage pulmonary resections pose challenges due to adhesions between the pulmonary parenchyma, chest wall, and hilum. This study aimed to investigate the surgical outcomes associated with Cavitron Ultrasonic Surgical Aspirator (CUSA) usage in thoracoscopic salvage pulmonary resections. Patients with centrally located advanced-stage lung tumors who underwent salvage anatomic resections following systemic or radiotherapy were included. They were categorized into CUSA and non-CUSA groups, and perioperative parameters and surgical outcomes were analyzed. Results: The study included 7 patients in the CUSA group and 15 in the non-CUSA group. Despite a longer median surgical time in the CUSA group (3.8 h vs. 6.0 h, *p* = 0.021), there was a significant reduction in blood loss (100 mL vs. 250 mL, *p* = 0.014). Multivariate analyses revealed that the use of CUSA and radiotherapy had opposing effects on blood loss (β: −296.7, 95% CI: −24.8 to −568.6, *p* = 0.034 and β: 282.9, 95% CI: 19.7 to 546.3, *p* = 0.037, respectively). In conclusion, while using CUSA in the salvage anatomic resection of centrally located lung cancer may result in a longer surgical time, it is crucial in minimizing blood loss during the procedure.

## 1. Introduction

According to the Taiwan Cancer Registry database 2010–2016, more than half of patients with lung cancer were diagnosed beyond stage IIIB [1]. These cases indicated systemic diseases potentially requiring additional systemic treatments according to the National Comprehensive Cancer Network (NCCN) guidelines, which recommend an assessment for operability only in patients with cancer stages below IIIA [2,3]. However, despite improvements in targeted therapy, immunotherapy, chemotherapy, and radiotherapy, local recurrence rates can be up to 35% and remain the leading cause of mortality. Several studies have reported that salvage thoracic surgery can lead to favorable prognoses in appropriately selected cases of individuals who have received systemic treatments [4,5,6,7].

Although lobectomy or limited resection can be smoothly performed for peripheral lesions, centrally located lung cancer with hilar node involvement frequently shows severe hilar adhesion or lung adhesion after pretreatment, which may influence the outcomes of the salvage operation [8,9,10,11,12,13]. Radiotherapy and systemic lung cancer treatment are also associated with fibrosis and unclear surgical planes, which increase the risks of postoperative vascular injuries and bronchopleural fistulas [10,14].

In thoracic surgery, multiple energy devices have been introduced for pneumolysis, hilar dissection, and mediastinal lymphadenectomy over the past two decades [15]. These include the LigaSure™ (Medtronic, Minneapolis, MN, USA) and the Harmonic Scalpel (Ethicon, a subsidiary of Johnson and Johnson, Cincinnati, OH, USA). Likewise, the Cavitron Ultrasonic Surgical Aspirator (CUSA) has gained popularity in liver surgeries due to its unique capability to vibrate and fragmentize hepatocytes while sparing blood vessels and bile ducts [16,17,18]. CUSA-assisted liver resection is widespread in abdominal surgery [19,20]. CUSA generates energy to disintegrate tissues with high water content, such as hepatocytes, soft lung tissue, and lymph nodes, while sparing tissues with lower water content, such as bile ducts, vascular walls, and the airway, enhancing its use for meticulous surgical hemostasis or ligation. In patients undergoing thoracic surgery, severe pulmonary parenchyma and hilar adhesion may develop after infection, radiation, or inflammation, necessitating extensive pneumolysis and hilar dissection for the subsequent pulmonary resection. Despite the availability of various energy devices suggested for aiding pneumolysis, their clinical effectiveness has shown potential for improvement [15]. However, a vessel-sparing approach with the Cavitron Ultrasonic Surgical Aspirator (CUSA) holds promise in facilitating salvage lung resection, particularly in centrally located lung cancer cases. This approach promotes the safety and standardization of hilar dissection during salvage surgeries within our practice. Hence, we present our experience applying the CUSA in assisting thoracoscopic salvage pulmonary resection.

## 2. Materials and Methods

### 2.1. Patient Inclusion

Patients with stage IIIB, C, and IV lung cancer who received treatment-intended salvage surgeries between 1 January 2017 and 31 December 2021 at a single medical center were reviewed retrospectively. Salvage surgery was defined as surgery for persistent or recurrent disease in patients initially treated with curative-intent non-operative management [21].

Inclusion criteria were patients with (1) age greater than 18 years; (2) lung cancer stage IIIB and above at diagnosis and treated with intention-to-cure chemotherapy, targeted therapy, immunotherapy, or radiotherapy; (3) centrally located primary tumor, defined by a distance of less than 2 cm between the tumor location and the proximal bronchial tree [22,23]; and (4) a history of salvage anatomical resection. Patients who did not undergo surgery with curative intent, such as those showing bronchopleural fistula, pneumothorax, persistent hemoptysis, and empyema thoracis, and those with peripheral primary tumors or incomplete data were excluded.

All patients were initially staged using computed tomography (CT) scans, positron emission tomography, and endobronchial ultrasound-guided transbronchial needle aspiration. The recommendations of our lung cancer board served as the basis for the protocols implemented for systemic therapy and radiotherapy. Only patients showing evidence of downstaging and low or moderate risk, according to the American College of Chest Physicians (ACCP) lung surgical risk algorithm [24], were eligible for surgery. The lung cancer board restaged and reviewed all patients before their operation.

All patients underwent video-assisted thoracoscopic surgery, which involved complete hilar dissection, lymph node dissection, and at least lobectomy of the affected lung site. The surgical procedure employed both LigaSure and a Laparascopic 5 mm Monopolar Hook (Addler, Mumbai, India) in all included patients. The decision to additionally use the CUSA during surgery was based on the surgeon’s preference and patient agreement. While the cost of the CUSA, approximately USD 200, was affordable for most patients, it may only be feasible for some, especially those who have experienced financial strain from prior treatments. The utilization of the CUSA was determined based on preoperative imaging that indicated severe adhesions involving major vital organs. In such cases, the CUSA was prepared and utilized when encountering challenging surgical planes during the operation. In cases where severe adhesion was observed in the hilar region or adjacent organs, the CUSA was employed as a primary tool for dissecting the appropriate surgical plane (Figure 1). The CUSA was meticulously applied around major vessels or organs, ensuring the cautious avoidance of direct contact with the target vessel or organ whenever possible. Its dissection depth of approximately 2 mm effectively removed cellular components while preserving delicate vascular structures. This method clearly identified the planned dissection plane between the fibrous tissue and the targeted structures. Any remaining fibrous tissue was subsequently dissected using alternative surgical instruments. LigaSure and staples were used for soft tissue, pulmonary vessel, and bronchus sealing and transection. Patient comorbidities were calculated using the Charlson comorbidity index [25]. Perioperative parameters and surgical outcomes, including blood loss, duration of hospitalization, the appearance of persistent air leaks, and overall complications, were analyzed.

### 2.2. Data Collection

The study’s primary outcome measure was blood loss, while the secondary outcomes included surgery-related complications, operation time, and duration of hospitalization. The total operation time was determined through measuring the duration from the skin incision to the completion of the wound closure. Surgery-related complications were categorized according to the Clavien–Dindo classification [26]. Persistent air leakage was defined by positive results in the air-leak test on postoperative day 7. The chest tube was kept in position until the patient demonstrated no air leaks and the daily volume of clear effusion was less than 200 mL.

### 2.3. Statistical Analysis

The significance of differences between the two groups was evaluated using Fisher’s exact test and the Mann–Whitney U test. Multivariate analysis for postoperative complications was performed with backward stepwise logistic regression; blood loss was calculated with backward stepwise linear regression. Statistical significance was defined as *p* < 0.05. All statistical analyses were performed with Statistical Package for Social Sciences (IBM SPSS Statistics, version 25) and Excel version 16.0 software (Microsoft Corp., Seattle, WA, USA).

This study was approved by the Institutional Review Board of Chung Shan Medical University Hospital (CS1-21185).

## 3. Results

### 3.1. Patient and Treatment Parameters in Salvage Anatomic Lung Resections with and without CUSA

A retrospective review was conducted on 22 patients between 2017 and 2021 after excluding 28 patients from the initial study population (Figure 2).

The demographic characteristics of the CUSA group are listed in Table 1. All patients in the CUSA group underwent lobectomy. Lobectomy was noted in 12 of 15 patients who underwent operations without the CUSA (two bilobectomies and one pneumonectomy, Table 1). None of the demographic factors differed significantly between the groups, which may be attributed to the small size of the study population. The median age of patients in the CUSA group was greater than that in the non-CUSA group (67 vs. 62 years). Both groups showed similar male smoker predominance and similar patterns for tumor differentiation and pathological type. The median Charlson comorbidity index was 5. More patients in the CUSA group had radiotherapy prior to the surgery (57.1% vs. 26.7%). All patients who had received radiotherapy before their operation showed an interval of at least six months between the operation and their last radiotherapy session. The median radiotherapy dosage in all patients was 4700 cGy (range, 3750–5500 cGy), and none had an accumulated dose >6000 cGy. The proportion of patients receiving immunotherapy in the CUSA group (42.9%) was more than twice that of the non-CUSA (20.0%). The corresponding proportions for chemotherapy were 71.4% and 93.3%, respectively, and those for targeted therapy were approximately 40%. The non-CUSA group included more ACCP low-risk patients (80.0% vs. 57.1%) and showed better pulmonary function. The pathological complete response (pCR) rate in the CUSA group (42.9%) was slightly more than twice that in the non-CUSA group (20.0%).

Both groups showed no cases of postoperative mortality in the 6-month follow-up. The data for surgical performance are presented in Table 2. Blood loss was significantly lesser in the CUSA group (median blood loss: 100 mL vs. 250 mL, *p* < 0.014). However, the duration of surgery in the CUSA group was longer than that in the non-CUSA group (median, 6.0 h vs. 3.8 h; *p* < 0.021). Although the intensive care unit (ICU) admission rate in the CUSA group (57.1%) was more than twice that in the non-CUSA (26.7%) group, both groups showed similar durations of ICU stay. Persistent air-leaks were the most frequent complication in our study; this complication was higher in the non-CUSA group (26.7% vs. 16.3%). Other surgical parameters, including duration of hospitalization, number of days of chest tube usage, and all complications, showed no statistically significant difference between groups. The Clavien–Dindo grade III complications were empyema thoracis and decortication in two patients of the non-CUSA group, in which one patient with acute respiratory distress syndrome after lobectomy required venous–venous extracorporeal membrane oxygenation support for five days.

### 3.2. Factors Affecting Complications in Thoracoscopic Salvage Anatomic Lung Resections

Univariate analysis for all complications revealed that age, radiotherapy, and non-adenocarcinoma pathology were associated with all complications. The factors with *p* < 0.5 were included in the multivariate analyses. Only age was a significant risk factor in the backward stepwise logistic regression analysis of all complications (odds ratio [OR] = 1.355; 95% confidence interval [CI], 1.065–1.724; Wald’s test *p*-value = 0.013; Table 3).

### 3.3. Factors Affecting Intraoperative Blood Loss in Thoracoscopic Salvage Anatomic Lung Resections

For blood loss, univariate linear regression identified CUSA application as the only factor showing a statistically significant association with blood loss (Table 4). The factors that showed *p* < 0.5 were included in the multivariate analyses. The backward stepwise linear regression model was statistically significant (F value = 3.968; *p* = 0.036, ANOVA). CUSA application and treatment with radiotherapy were risk factors associated with intraoperative blood loss, with β(95% CI) of −296.7 (−24.8 to −568.6, *p* = 0.034) and 282.9 (19.7 to 546.3, *p* = 0.037), respectively. Furthermore, concerning significant blood loss, three patients in the non-CUSA group encountered excessive intraoperative blood loss exceeding 500 mL (500, 550, and 1500 mL, respectively).

## 4. Discussion

This investigation is the first documented application of the CUSA in lung resection in the English-published medical literature. The findings of this study suggest that incorporating the CUSA into surgical procedures can contribute to a decrease in blood loss and a reduction in significant complications during salvage lung resections for centrally located tumors. In this study, a total of 22 patients with centrally located lung cancer were identified. The majority of patients shared common characteristics such as being male, smokers, and having non-adenocarcinoma, which aligns with the demographic profile reported in the existing literature [27]. The median patient age was approximately 65 years, and the median Charlson comorbidity index was 5. Surgery was performed only for patients with an appropriate ACCP risk classification and adequate pulmonary function.

Among the various modalities for oncological treatment, chemotherapy was the most popular in the non-CUSA group and accounted for 21.9% more cases in this group. In contrast, immunotherapy and radiotherapy accounted for approximately 20% more cases in the CUSA group. The results showed that the pathological complete response (pCR) rate was approximately 22.9% higher in the CUSA group. Among the six patients who achieved pCR, five had undergone radiotherapy, four had received immunotherapy, and all six had received chemotherapy. Although patients showing pCR had better overall survival, the post-treatment hilar structure was difficult to approach due to calcification, fibrosis, and adhesion [28,29]. In cases involving prolonged radiotherapy and pCR, despite using other energy devices, adding the CUSA could further reduce the blood loss from a median value of 250 mL to 100 mL, which was calculated to be 296.7 mL with linear regression. Moreover, despite older patients with higher ACCP risk stratification and poorer pulmonary function, the CUSA group showed fewer air leaks and no cases with severe complications (Clavien–Dindo grade ≥ III).

Salvage lung resections are challenging procedures associated with high rates of surgical morbidity or even mortality. Thus, a simple, straightforward tool for these procedures is desired, and the CUSA could address this requirement. Centrally located lung cancer with a massive tumor burden is challenging for achieving appropriate surgical traction and safe dissection. Considering the difficulty of sparing hilar vessels, the blood loss in salvage pulmonary surgery ranges from 0 to 4400 mL [21]. The CUSA can be used for structural dissection in cases showing adhesion around the hilar and mediastinum. Some fibrotic lymph nodes formed post-treatment were treated with CUSA to clarify the safety border of vessels and airways and were removed successfully. The more straightforward clearance of the surgical plane from the inflammatory fibrotic chaos provides a safety guarantee. Intraoperative blood loss during liver parenchyma resection is known to be associated with postoperative complications and oncological outcomes [30,31]. The application of the CUSA was first described by Hodgson et al. in 1979 and popularized in neurosurgery and hepatic surgery since 1980 [17]. Some of the literature in the early 1990s revealed that CUSA application in major liver resection caused less blood loss and necessitated lower amounts of transfusion [32,33]. Even in minimally invasive surgery, using the CUSA for major liver resection is an important tool for surgeons [34]. As for pulmonary resection, we demonstrated that even in less favorable surgical conditions, the CUSA group showed less than half the median blood loss in the non-CUSA group, equivalent to approximately one unit (250 mL) of blood. The CUSA not only avoided injury to the major vessel but also prevented lung parenchyma injury and neo-vascular structure avulsion during dissection and minimized blood loss.

The goal of radiotherapy in these patients with advanced lung cancer was to improve patient survival via direct action on the tumor, allowing downstaging and complete resection of tumors initially considered inoperable or even pCR. However, in patients with centrally located lung cancer, direct action on the hilar site may result in additional adhesion and fibrosis of hilar structures, rendering these structures more fragile. Furthermore, radiotherapy damages the blood vessels and the perivascular tissues surrounding the area, resulting in brittle blood vessels that can often break during surgery. Additionally, the tissues surrounding the treated area may become inflamed and swollen, potentially increasing the risk of blood loss during surgery. Several studies have shown that radiotherapy is a risk factor for blood loss in diseases such as head and neck cancer [35,36]. For advanced lung cancer, patients who underwent lung lobectomy after radiotherapy had a higher risk of bleeding and a higher complication rate than those who had not received radiotherapy [37,38]. The interval between surgery and radiotherapy is essential: a longer interval may theoretically lead to much greater fibrosis development, causing more blood loss and morbidities. The interval between the last radiotherapy and surgery differed across studies, ranging from 18 weeks to 96 weeks [5,11,29,39]. In our study, the surgical intervention was performed at least six months after the last radiotherapy. Even though the proportion of patients who had received radiotherapy was twice as high in the CUSA group compared to the non-CUSA group, the former still exhibited lower blood loss. This study highlights the feasibility of salvage lung surgeries utilizing the CUSA and other energy devices in an experienced thoracic surgical center with appropriately selected patients. The approach demonstrates reasonable morbidity and mortality rates. No surgery-related mortalities were observed in our study. Previous studies demonstrated that surgical mortality was mainly associated with the therapy before the operation. For patients undergoing definitive chemoradiotherapy before surgery, Bograd et al. observed a mortality rate of 6.7% in their series, which could be attributable to surgical complexity [40]. Another study by Antonoff et al. reported a mortality rate of 4.8% [41]. However, in patients who received only neoadjuvant and targeted therapy, the surgery-related mortality rate was nearly 0% [42,43].

Postoperative complications developed in 40.9% of the patients in our study, including six in the non-CUSA group and three in the CUSA group, without showing a statistically significant difference between groups. Some studies have reported complication rates of 30–57% [21,40]. In our study, age was an independent risk factor associated with all complications in multivariate logistic regression. The patients showing complications in our study were much older (70 years old vs. 58 years old; OR = 1.355; *p* = 0.013). Several other studies have reported similar findings [44,45]. Elderly patients are much more fragile and may have several cardiopulmonary comorbidities after chemotherapy, targeted therapy, and radiotherapy. Thus, salvage surgery in these patients should be carefully selected, considering both oncological and quality-of-life benefits. Although patients in the CUSA group were five years older than those in the non-CUSA group and the APCC risk stratification was higher, the complication rate in the CUSA group was not higher. In contrast, complications requiring surgical intervention only occurred in the non-CUSA group.

Incidental bronchial injuries represent a severe additional complication that can occur during salvage lung surgery. Hamada et al. reported that the incidence of bronchial pleural fistula after salvage operation was approximately 2% in their study, while Bograd et al. reported that more than 80% of the patients required preventive bronchial stump flap coverage in their series [10,40]. In contrast, the patients in our series did not undergo preventive stump coverage, and all of our patients were treated with an endovascular gastrointestinal anastomosis stapler (Endo GIA; Medtronic, Minneapolis, MN, USA) for dividing and sealing the bronchus. Most bronchial stump ligation procedures were performed with the Thick Tri-Staple Reload (Medtronic, Purple). However, in cases where the stapler application was hindered by fibrosis and adhesions, Endo GIA™ Black Reload with Tri-Staple™ was used instead. The CUSA allowed for the easy dissection of the airways from the fibrotic tissue and clearly reduced the chances of bronchus injury, potentially preserving more bronchial feeding vessels and avoiding major complications.

Approximately half of the patients in the CUSA group experienced persistent air leaks. Although the statistical analysis did not yield significant results, these findings suggest a potential decrease in the occurrence of air leaks. Air leaks were observed in patients with deep lung or airway injuries. Using CUSA in dissecting the lung parenchyma and surrounding adhesion tissue while avoiding penetration through the parietal pleural layer may contribute to a lower incidence of persistent air leaks and reduced blood loss.

One of the major factors advocating against the widespread adoption of the CUSA is efficiency. In our study, the median operation time in all patients was 4.2 h (range, 3.5–5.0 h), similar to findings from other investigations [21,46]. However, the operation time in the CUSA group was significantly longer (6.0 h vs. 3.8 h, *p* = 0.021). The dissection process using the CUSA was characterized by its slow pace, with a progression of approximately 1–2 mm per minute. At times, the device required cooling down, and once a clear plane was reached, the remaining fibrous or tiny vascular tissue had to be treated using other devices. Reverting to the CUSA after switching devices took approximately 5–10 min for the machine to rewarm [47]. As a result, the CUSA was typically reserved for surgeries anticipated to be time-consuming.

This study had several limitations. First, it is an observational study, and the choice of CUSA was based on the surgeon’s preferences and requirements. The two groups were not matched; thus, the CUSA group may have had more fibrotic adhesions. We acknowledge that bias towards the CUSA was present in our study, particularly in more challenging cases, such as those involving post-radiotherapy patients, elderly individuals, and cases with significant hilar adhesion. At our institution, the selection of using the CUSA was not randomized, and only four out of our eight attending surgeons had experience with its utilization. Second, the case numbers in both groups were small since patients were selected from a single institution after applying criteria for lung function and performance status. Thus, the results may not have shown statistical significance. Third, several literature reviews have achieved no consensus on the definition of centrally localized lung cancer [48]. Based on the Radiation Therapy Oncology Group definition, we defined central lesions in cases wherein the distance between the tumor’s location and the proximal bronchial tree was less than 2 cm. Fourth, the heterogeneity in treatment prior to the operation may have resulted in differences in surgical outcomes. Chemoimmunotherapy with programmed death (PD)-1 inhibitors have been reported to be associated with a pCR rate of 15% to 40% in a previous literature review [49,50,51]. In contrast, preoperative tyrosine kinase inhibitor induction therapy was associated with a pCR rate of 12.1% and a major pathological response rate of 24.2% [42,52,53]. The variations in pathological response could have contributed to discrepancies in the surgical scenario and complications observed.

## 5. Conclusions

In our pioneering study, despite being older in age and having a higher APPC risk stratification, poorer pulmonary function, and a higher frequency of receiving radiotherapy, patients who utilized the CUSA demonstrated reduced blood loss, potentially lower rates of persistent air leaks, and fewer severe complications during post-cure treatment-intent anatomic lung resection for advanced centrally located lung cancer. The CUSA provides a simple and safe approach for avoiding injuries to essential structures and facilitating hilar dissection for salvage lung surgeries.

## Figures and Tables

**Figure 1 cancers-15-04069-f001:**
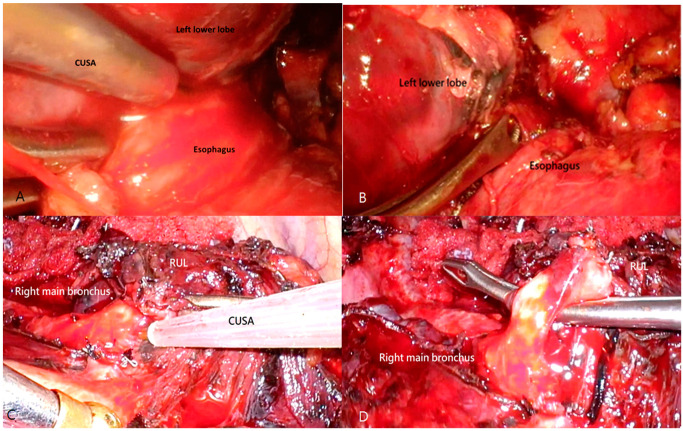
The CUSA was used to safely dissect the right upper lobe (RUL) bronchus, embedded within dense, fibrotic tissue of high firmness. Surgical steps in applying CUSA for salvage lobectomy: After radiotherapy, the whole esophagus was adherent to the left lower lobe, and we were initially unaware that it was the esophagus. The CUSA was used to create the surgical plane first (**A**). After clarifying the definitive surgical plane, the left lower lobe and esophagus were separated without injury to the lung or esophagus, and the esophagus was revealed clearly (**B**). After chemotherapy and targeted therapy, the right main bronchus and RUL showed dense fibrosis and were encased with fibrotic lymph nodes (**C**). The route for looping around the right upper lobe bronchus could not be identified (**D**). After the application of the CUSA for fibrosis division and lymph node harvesting, the surgical plane was created.

**Figure 2 cancers-15-04069-f002:**
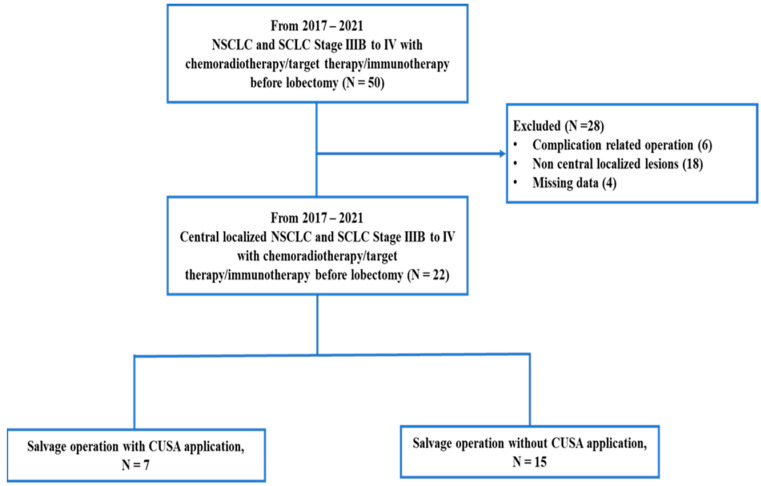
The recruitment algorithm.

**Table 1 cancers-15-04069-t001:** Characteristics of patients who received salvage pulmonary surgery with and without CUSA.

	Non-CUSA	CUSA	*p*
	*N* = 15	*N* = 7	
Age, median (IQR)	62 (56–69)	67 (58–74)	0.447
Sex			
Female	3 (20.0)	2 (28.6)	0.739
Male	12 (80.0)	5 (71.4)	
Morbidity score	5.0 (3–9)	5.0 (4–8)	0.837
Smokers	10 (67.7)	5 (71.4)	0.999
Pathology			0.227
Adenocarcinoma	7 (46.7)	3 (42.9)	
Squamous cell carcinoma	5 (33.3)	2 (28.6)	
Others	3 (20.0)	2 (28.6)	
Differentiation			0.999
Moderately differentiated	10 (66.7)	4 (57.1)	
Poorly differentiated	4 (26.7)	2 (28.6)	
Tumor size (mm, IQR)	40.0 (30–70)	44.0 (28–60)	0.891
Stage			0.273
IIIB+C	9 (60.0)	5 (71.4)	
IV	6 (40.0)	2 (28.6)	
FEV1, Median (IQR)	84.0 (73.0–89.0)	71.0 (58.0–100.0)	0.332
DLCO, Median (IQR)	75.0 (65.0–90.0)	71.0 (55.0–79.0)	0.298
ACCP risk grade			0.334
Low risk	12 (80.0)	4 (57.1)	
Moderate risk	3 (20.0)	3 (42.9)	
Immunotherapy	3 (20.0)	3 (42.9)	0.334
Targeted therapy	6 (40.0)	3 (42.9)	0.999
Radiotherapy	4 (26.7)	4 (57.1)	0.343
Chemotherapy	14 (93.3)	5 (71.4)	0.227
Operation			0.445
Lobectomy	12 (80.0)	7 (100)	
Bilobectomy	2 (13.3)	0 (0.0)	
Pneumonectomy	1 (6.7)	0 (0.0)	
pCR	3 (20.0)	3 (42.9)	0.334

IQR, interquartile range; pCR, pathological complete response; ACCP, American College of Chest Physician lung surgical risk classification; FEV1, forced expiratory volume in 1 s; DLCO, diffusion capacity of the lungs for carbon dioxide; morbidity score: Charlson comorbidity index.

**Table 2 cancers-15-04069-t002:** Surgical parameters of advanced lung cancer patients who underwent salvage resection.

ALL Type	Non-CUSA	CUSA	*p*-Value
	*N* = 15	*N* = 7	
Blood loss, median (IQR)	250 (150–450)	100 (100–200)	0.014 *
Surgical duration, median (IQR)	3.8 (3.5–4.5)	6.0 (4.0–7.0)	0.021 *
Duration of hospitalization, days, median (IQR)	9.0 (8.0–12.0)	11 (8–13)	0.731
ICU admission	4.0 (26.7)	4.0 (57.1)	0.343
Length of ICU admission (days), median (IQR)	2.0 (1.0–5.0)	2.5 (1.5–5.5)	0.368
Period of chest tube insertion, days, median (IQR)	6.0 (5.0–9.0)	7.0 (4.0–9.0)	>0.999
Persistent air leak, n (%)	4.0 (26.7)	1.0 (14.3)	>0.999
All complications	6.0 (40.0)	3.0 (42.9)	>0.999
Grade III complications	2.0 (13.3)	0.0 (0.0)	>0.999

* *p* < 0.05; ICU, intensive care unit; IQR, interquartile range.

**Table 3 cancers-15-04069-t003:** Factors affecting complications in patients with advanced lung cancer undergoing salvage surgery.

Complication		With	Without	*p*
		*N* = 9 (100)	*N* = 13 (100)	
Sex				
Male	n (%)	7 (77.8)	10 (76.9)	0.962
Female	n (%)	2 (22.2)	3 (23.1)	
Age	median (IQR)	70 (68, 73)	58 (56, 62)	0.001 *
	OR (95% CI)	1.355	(1.065, 1.724)	0.013 **
Smoker	n (%)	7 (77.8)	8 (61.5)	0.648
Charlson comorbidity index	median (IQR)	8 (5, 9)	4(3, 8)	0.071
ACCP low risk	n (%)	5 (55.6)	11 (84.6)	0.178
Pathology				0.021 *
Adenocarcinoma	n (%)	1 (11.1)	9 (69.2)	
Squamous cell carcinoma	n (%)	4 (44.4)	3 (23.0)	
Others	n (%)	4 (44.4)	1 (7.7)	
Tumor stage				
IIIB+C	n (%)	5 (55.6)	9 (69.2)	0.147
IV	n (%)	4 (44.4)	4 (3.08)	
Tumor size	n (%)	49 (40, 70)	39(30, 45)	0.235
Radiotherapy	n (%)	6 (66.7)	2 (15.4)	0.026 *
Chemotherapy	n (%)	9 (100)	10 (76.9)	0.204
Target Therapy	n (%)	2 (22.2)	7 (53.8)	0.203
Immunotherapy	n (%)	4 (44.4)	2 (15.4)	0.178
CUSA	n (%)	3 (33.3)	4 (30.8)	0.899

* *p* < 0.05, univariate analysis; ** *p* < 0.05, backward stepwise logistic regression; ACCP, American College of Chest Physicians lung surgical risk classification; IQR, interquartile range; OR, odds ratio; CI, confidence interval.

**Table 4 cancers-15-04069-t004:** Factors affecting blood loss in patients undergoing salvage surgery.

			Blood Loss, mL	
		n	Median (IQR)	*p*
Age	year	22	−210.4 (−494.4, 73.5) †	0.583
Gender	male	17	150 (150, 350)	0.401
	female	5	100 (100, 350)	
Smoker	yes	15	200 (150, 350)	0.237
	no	7	150 (100, 250)	
ACCP	Low	16	150 (125, 135)	0.999
	Moderate	6	175 (100, 350)	
Morbidity score		22	37.8 (−6.9, 82.4) †	0.093
Tumor size	mm	22	0.06 (−7.1, 7.3) †	0.121
Tumor stage	IIIB+C	14	150 (100, 350)	0.224
	IV	8	225 (200, 250)	
Pathology	AdenoCa	10	150 (100, 350)	0.611
	SCC	7	150 (150, 350)	
	Others	5	350 (150, 350)	
Radiotherapy	yes	8	275 (200, 350)	0.330
	no	14	150 (150, 250)	
			282.9 (19.7, 546.3) †,‡	0.037 **
Chemotherapy	yes	19	150 (100, 350)	0.160
	no	3	100 (50, 250)	
Target therapy	yes	9	150 (100, 250)	0.556
	no	13	150 (150, 350)	
Immunotherapy	yes	6	250 (150, 350)	0.910
	no	16	150 (150, 350)	
CUSA	yes	7	100 (100, 200)	0.014 *
	no	15	250 (150, 450)	
			−296.7 (−568.6, −24.8) †,‡	0.034 **
pCR	yes	6	150 (100, 350)	0.541
	no	16	175 (125, 400)	

SCC, squamous cell carcinoma, AdenoCa, adenocarcinoma; pCR, pathological complete response; IQR, interquartile range; ACCP, American College of Chest Physicians lung surgical risk classification. * *p* < 0.05. ** *p* < 0.05, backward stepwise linear regression. † go for linear regression. ‡ β (95% CI.).

## Data Availability

The data presented in this study are available on request from the corresponding author. The data are not publicly available due to restrictions posed by the institution.

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
