# Peer review of "Reduction of Blood Loss by Means of the Cavitron Ultrasonic Surgical Aspirator for Thoracoscopic Salvage Anatomic Lung Resections"

_cancers, 2023, doi:10.3390/cancers15164069_

Round 1

Reviewer 1 Report

I would like to congratulate you on the study you presented. It is thoroughly developed, and well supported by figures and tables. I found your proposal very interesting And I think I'll start using your method as soon as possible

Author Response

We truly appreciate your positive feedback on the study and the manuscript. It is gratifying to know that you found the content interesting and well-supported by figures and tables.

Reviewer 2 Report

 This paper is reviewing the use of CUSA as an alternative to LigaSure or other electro- cauterization devices. It is a good alternative from the perspective of blood loss. The study compares patients in whom CUSA was used with non-CUSA. The advantages include decreased blood loss, smaller risk of bronchial injury and a lower incidence of persistent air leaks. The current study shows a statistically significant reduction in blood loss, but further studies on many cases are required.

The paper is well structured, the statistical data is appropriate, and the references are up to date.

Still, I consider that saying “other electro-cauterization devices” should be replaced with a specific name.

Otherwise, the paper can be published in the current form.

Author Response

We appreciate your positive feedback, specifically on the structure, statistical data, and references.

In response to your comments on specifying the “other electro-cauterization devices”, we made corrections in the Materials and Methods section and highlighted the changes in red as follows:

All patients underwent video-assisted thoracoscopic surgery, which involved complete hilar dissection, lymph node dissection, and at least lobectomy of the affected lung site. The surgical procedure employed both LigaSure and electro-cauterization devices Laparascopic 5mm Monopolar Hook (Addler, Mumbai, India) in all included patients.

Reviewer 3 Report

Dear Editor and Authors,

It was my pleasure to evaluate this manuscript titled “Reduction of Blood Loss by Cavitron Ultrasonic Surgical Aspirator for Thoracoscopic Salvage Anatomic Lung Resections” by Dr. Wang  and colleagues from Chung Shan Medical University Hospital from Taiwan.

In this study the authors offer their experience utilizing the CUSA devise to dissect fibrotic and tough tissues in patients with lung cancer undergoing salvage surgery. This is an interesting study and having used the device personally I think this subject is appealing to the thoracic surgical community.

There are naturally some limitations to this study such as its retrospective nature, its single institution origin and its small sample size. However, it utilizes sound methodology, robust exclusion and inclusion criteria, a solid statistical analysis and it is well presented with informative tables and figures in clear and understandable language.

I do have some minor comments to offer:

Comments:

1.       Minor Comment: No affiliations are listed despite author names been numbered. Am I missing something or is this an omission?

2.       I disagree that “Despite the availability of various energy devices suggested for aiding pneumolysis, their clinical effectiveness has been unsatisfactory” – line 52. Both the LigaSure and the Harmonic are excellent energy devices widely used and despite a different mechanism of function also offer excellent results. Therefore, I suggest the authors re-phrase this.

3.       One issue I would like addressed is the fact that the use of CUSA was dependent of surgeon’s preference. This as you can understand raises the very likely possibility of bias!! The authors have addressed in their limitation section but is there a way to statistically see if the distribution of surgeons using the CUSA was random? i.e. Did all surgeons use the CUSA but they differed in which patients they would use it or only 1 or 2 used it and the rest did not?

4.       I suggest that in the Material and Method’s section a different subheading of Statistical Analysis be added with the methodology used (lines 121 - 127).

5.       I am not sure that table 1 is needed. I think table 2 is more than adequate to present the demographics and I don’t see the point of table 1!

6.       Given the small sample size available I believe that the multivariable models (blood loss and complications) are overloaded because the authors have included a large number of variables (I counted 12 in total such as age, female, smoker, Charlson co-morbidity….CUSA) in them.

7.       Rephrase the title of Table 5. It makes no sense. I would suggest “Factor affecting Blood Loss in Patients Undergoing Salvage Surgery”.

In conclusion, this is a nice little study which I feel has merit to be presented in the literature following some minor correcting/editing. Wishing well to all.

Minor language editing required.

Author Response

Point-by-point response to comments by Reviewer 3

Dear Editor and Authors,

It was my pleasure to evaluate this manuscript titled “Reduction of Blood Loss by Cavitron Ultrasonic Surgical Aspirator for Thoracoscopic Salvage Anatomic Lung Resections” by Dr. Wang and colleagues from Chung Shan Medical University Hospital from Taiwan.

In this study the authors offer their experience utilizing the CUSA devise to dissect fibrotic and tough tissues in patients with lung cancer undergoing salvage surgery. This is an interesting study and having used the device personally I think this subject is appealing to the thoracic surgical community.

There are naturally some limitations to this study such as its retrospective nature, its single institution origin and its small sample size. However, it utilizes sound methodology, robust exclusion and inclusion criteria, a solid statistical analysis and it is well presented with informative tables and figures in clear and understandable language.

I do have some minor comments to offer:

Comments:

  1. Minor Comment: No affiliations are listed despite author names been numbered. Am I missing something or is this an omission?

Response 1: This is an omission to conceal the authors’ identities for the blinded review process. We edited the revised manuscript by adding back the authors’ affiliations.

  1. I disagree that “Despite the availability of various energy devices suggested for aiding pneumolysis, their clinical effectiveness has been unsatisfactory” – line 52. Both the LigaSure and the Harmonic are excellent energy devices widely used and despite a different mechanism of function also offer excellent results. Therefore, I suggest the authors re-phrase this.

Response 2: We would like to thank the reviewer for this very insightful comment. Indeed, we used LigaSure in all of our cases, with/without CUSA. We re-phrased it to: “Despite the availability of various energy devices suggested for aiding pneumolysis, their clinical effectiveness has shown potential for improvement.”

  1. One issue I would like addressed is the fact that the use of CUSA was dependent of surgeon’s preference. This as you can understand raises the very likely possibility of bias!! The authors have addressed in their limitation section but is there a way to statistically see if the distribution of surgeons using the CUSA was random? i.e. Did all surgeons use the CUSA but they differed in which patients they would use it or only 1 or 2 used it and the rest did not?

Response 3: We described the timing of the use of CUSA in the last paragraph of M&M as:

“The utilization of CUSA was determined based on preoperative imaging that

indicated severe adhesions involving major vital organs. In such cases, CUSA was

prepared and utilized when encountering challenging surgical planes during the

operation. In cases where severe adhesion was observed in the hilar region or

adjacent organs, the CUSA was employed as a primary tool for dissecting the

appropriate surgical plane:” To further clarify various surgeons’ involvement, we added in the limitation section: “We acknowledge that bias towards the CUSA was present in our study, particularly in more challenging cases, such as those involving post-radiotherapy patients, elderly individuals, and cases with significant hilar adhesion. At our institution, the selection of using CUSA was not randomized, and only four out of our eight attending surgeons had experience with its utilization.”

  1. I suggest that in the Material and Method’s section a different subheading of Statistical Analysis be added with the methodology used (lines 121 - 127).

Response 4: Thank you for your suggestion. We used different subheadings: 2.1 Patient inclusion, 2.2 Data collection, and 2.3 Statistical analysis, accordingly.

  1. I am not sure that table 1 is needed. I think table 2 is more than adequate to present the demographics and I don’t see the point of table 1!

Response 5: We agreed with your view on Table 1 and removed Table 1 from the manuscript.

  1. Given the small sample size available I believe that the multivariable models (blood loss and complications) are overloaded because the authors have included a large number of variables (I counted 12 in total such as age, female, smoker, Charlson co-morbidity….CUSA) in them.

Response 6: Thank you for your thoughtful comment. We would like to stress the importance of considering these factors and hope to reflect this in our statistical analysis.

  1. Rephrase the title of Table 5. It makes no sense. I would suggest “Factor affecting Blood Loss in Patients Undergoing Salvage Surgery”.

Response 7: We amended the title of Table 5 according to your suggestion.

In conclusion, this is a nice little study which I feel has merit to be presented in the literature following some minor correcting/editing. Wishing well to all.
